# Association between Organizational Support and Turnover Intention in Nurses: A Systematic Review and Meta-Analysis

**DOI:** 10.3390/healthcare12030291

**Published:** 2024-01-23

**Authors:** Petros Galanis, Ioannis Moisoglou, Ioanna V. Papathanasiou, Maria Malliarou, Aglaia Katsiroumpa, Irene Vraka, Olga Siskou, Olympia Konstantakopoulou, Daphne Kaitelidou

**Affiliations:** 1Clinical Epidemiology Laboratory, Faculty of Nursing, National and Kapodistrian University of Athens, 11527 Athens, Greece; pegalan@nurs.uoa.gr (P.G.); aglaiakat@nurs.uoa.gr (A.K.); 2Faculty of Nursing, University of Thessaly, 41500 Larisa, Greece; iomoysoglou@uth.gr (I.M.); malliarou@uth.gr (M.M.); 3Department of Radiology, P. & A. Kyriakou Children’s Hospital, 11527 Athens, Greece; irenevraka@yahoo.gr; 4Department of Tourism Studies, University of Piraeus, 18534 Piraeus, Greece; olsiskou@nurs.uoa.gr; 5Center for Health Services Management and Evaluation, Faculty of Nursing, National and Kapodistrian University of Athens, 11527 Athens, Greece; olympiak1982@hotmail.com (O.K.); dkaitelid@nurs.uoa.gr (D.K.)

**Keywords:** organizational support, turnover intention, nurses, systematic review, meta-analysis

## Abstract

Although recent studies suggest a negative relationship between organizational support and turnover intention among nurses, there has been no systematic review on this issue. The aim of this systematic review and meta-analysis was to synthesize and evaluate the association between organizational support and turnover intention in nurses. The review protocol was registered with PROSPERO (CRD42023447109). A total of eight studies with 5754 nurses were included. All studies were cross-sectional and were conducted after 2010. Quality was moderate in five studies and good in three studies. We found a moderate negative correlation between organizational support and turnover intention since the pooled correlation coefficient was −0.32 (95% confidence interval: −0.42 to −0.21). All studies found a negative correlation between organizational support and turnover intention ranging from −0.10 to −0.51. A leave-one-out sensitivity analysis showed that our results were stable when each study was excluded. Egger’s test and funnel plot suggested the absence of publication bias in the eight studies. Subgroup analysis showed that the negative correlation between organizational support and turnover intention was stronger in studies in China and Australia than those in Europe. Organizational support has a moderate negative correlation with turnover intention in nurses. However, data regarding the impact of organizational support on turnover intention are limited. Moreover, our study had several limitations, and thus, we cannot generalize our results. Therefore, further studies should be conducted to assess the independent effect of organizational support on turnover intention in a more valid way. In any case, nursing managers should draw attention to organizational support by developing effective clinical practice guidelines for nurses so as to reduce turnover intention.

## 1. Introduction

Nurses, as frontline healthcare workers, are at the core of patient care delivery. They provide patients with the majority of care during their hospital stay, ensure the quality and safety of care, and contribute to patient satisfaction with it [1,2,3]. Ensuring the necessary resources for nursing staff is an essential prerequisite for providing quality nursing care [4]. However, over time, nurses’ work environments have been characterized by nursing understaffing and inadequate organizational support [5,6]. The consequence of all the above is the occurrence of burnout among nurses, their lack of work engagement, and their intention to leave their profession [5,7,8]. The pandemic period of COVID-19 found healthcare systems struggling with the same organizational problems and weaknesses as the pre-COVID period [9]. During this period, the high workload and work intensity further burdened nurses, who were more likely to declare their intention to leave the profession [10].

Nurses’ turnover constitutes, over time, a phenomenon that characterizes their profession. Nurses’ turnover can be defined as voluntarily leaving a particular position and moving to another within the same organization or to another healthcare organization, or ultimately leaving the profession and choosing another profession [11]. The prevalence of nurses declaring their turnover intention was high before the pandemic, reaching over 40% [12], and remained high during the pandemic period [13]. Among healthcare workers, nurses report the highest intent to leave the job rate [14]. The main factors related to nurses’ turnover intention are their working environment and, in particular, nursing staffing and the adequacy of resources [15,16]; stress related to work, to constant contact with patients and their relatives, and to conflicts with colleagues and supervisors [17]; organizational culture and fatigue [18]; shift work and organizational commitment [19]; and job dissatisfaction, burnout and depression [20,21,22].

After the COVID-19 pandemic, many nurses chose to stay in their jobs as it was difficult to change jobs due to the loss of many vacancies as an effect of the pandemic. As working conditions still remain difficult and nurses experience high rates of dissatisfaction and burnout [23], they choose quiet quitting, which is characterized by a decrease in their performance [24,25]. However, those who choose quiet quitting also report a high percentage of turnover intention [26]. Therefore, even the option of quiet quitting, which is a kind of defensive attitude of self-preservation for nurses in the demanding working environment, is not able to stop the tendency of nurses to leave the profession. For the factors associated with turnover intention, immediate solutions should be sought by the administrations of healthcare organizations worldwide as turnover intention is a strong determinant of actual turnover behavior [27,28].

Within the demanding and challenging work environment of healthcare delivery, a crucial factor influencing the turnover intention of nurses is the perceived organizational support they receive. According to the theory of perceived organizational support, employees believe that their work organization values their contribution and cares about their well-being [29]. In particular, perceived organizational support consists of organizational rewards, favorable job conditions, assistance to an employee to perform tasks efficiently and manage stressful situations, and support from the supervisor [30].

The benefits of organizational support are multifaceted, affecting nurses and their performance. When nurses receive organizational support, work engagement increases [7], nurses’ innovative behavior is enhanced [31], they report greater affective commitment [32], the quality of care is improved, and nurses experience higher job satisfaction, psychological well-being, and lower burnout and anxiety [33,34,35]. The degree of organizational support received by nurses influences their intention to stay in the profession [36,37,38,39]. As there are already significant shortages of nurses worldwide, which are projected to continue until 2030 [40], halting the turnover phenomenon will help towards the availability of nurses and better staffing of healthcare services. Therefore, organizational support is an important tool for achieving this objective.

Among other organizational factors, recent studies found a negative relationship between organizational support and turnover intention among nurses [36,41,42]. However, to date, no systematic review has been published on the association between organizational support and turnover intention. Thus, the aim of this systematic review and meta-analysis was to synthesize and evaluate the relationship between organizational support and turnover intention among nurses.

## 2. Materials and Methods

The review protocol was registered with PROSPERO (CRD42023447109).

### 2.1. Search Methods

We searched PubMed, Medline, Scopus, Cinahl, Web of Science, and Cochrane from inception to 21 August 2023. We searched in all fields using the following strategy: ((nurses OR nursing OR nurse OR “nursing staff”) AND (“organizational support” OR “organisational support”)) AND (“turnover intention” OR intention OR “intent to leave” OR turnover OR “intent to quit” OR “intention to leave” OR “intention to quit”). The duration of the literature search of the studies by the authors lasted from 14 to 21 August 2023.

### 2.2. Selection Process

Our inclusion criteria were the following: (a) studies that included nurses working in clinical settings, (b) articles published in English, (c) studies that investigated the relationship between organizational support and turnover intention in nurses, and (d) studies that used valid instruments to measure organizational support and turnover intention. Organizational support is a broad term that can vary across different organizations and countries. In our review, we included studies that measured the perceived organizational support among nurses. In particular, perceived organizational support was defined as comprising nurses’ overall perceptions and beliefs about how much organizations value and respect nurses’ well-being and job satisfaction. We excluded meeting or conference abstracts, case reports, qualitative studies, reviews, meta-analyses, protocols, editorials, and letters to the Editor. Moreover, we excluded studies that measured nurses’ intention to stay instead of intention to leave. Additionally, we excluded studies that simultaneously included nurses and other healthcare workers, so it was impossible to extract results only for nurses.

Applying the inclusion and exclusion criteria, two independent authors screened titles and abstracts of the records. Then, they screened the full texts of the records. A third senior author resolved all disagreements between the two independent authors.

### 2.3. Quality Appraisal

We used the Joanna Briggs Institute critical appraisal tools to assess the quality of studies included in our review [43]. All studies in our review were cross-sectional, and thus, we employed the Joanna Briggs Institute critical appraisal tool for this type of study. In particular, the Joanna Briggs Institute tool for cross-sectional studies comprises eight items. Higher scores indicate better quality. In particular, ≤3 is considered as having high-risk bias, 4–6 as having moderate-risk bias, and 7–8 as having low-risk bias. Two scholars performed the bias assessment.

### 2.4. Data Abstraction

Two scholars independently extracted the following data from each study: first author, year of publication, country, data collection time, percentage of females, age, sample size, study design, sampling method, clinical settings, assessment tools for organizational support and turnover intention, response rate, correlation coefficient between organizational support and turnover intention, unstandardized coefficient beta from linear regression models with turnover intention as the dependent variable, and *p*-values.

### 2.5. Synthesis

All studies presented correlation coefficients between organizational support and turnover intention while only two studies presented unstandardized coefficient betas. Thus, we performed meta-analysis for the correlation coefficients and not for unstandardized coefficient betas. In particular, we calculated the pooled correlation coefficient between organizational support and turnover intention and the 95% confidence interval (CI). Correlation coefficient between −0.1 and −0.29 indicates a small effect, between −0.3 and −0.49, a moderate effect, and higher than 0.49, a large effect [44]. Additionally, we assessed heterogeneity between studies by calculating the I^2^ statistics and the *p*-value for the Hedges Q statistic. I^2^ values higher than 75% indicate high heterogeneity while a *p*-value < 0.1 for the Hedges Q statistic indicates statistically significant heterogeneity [45]. Heterogeneity between studies was high, and thus, we applied the random effects model to calculate the pooled correlation coefficient. A leave-one-out sensitivity analysis was employed to estimate the influence of each study on the pooled correlation coefficient. A priori, we considered country, data collection time, percentage of females, sample size, quality of studies, and response rate as sources of heterogeneity. To examine heterogeneity, we performed subgroup analysis for categorical variables and meta-regression for continuous variables. We used Egger’s test and funnel plot to estimate publication bias [46]. *p*-value < 0.05 for Egger’s test and asymmetry of funnel plot indicate the presence of publication bias. We used OpenMeta [Analyst] to perform the meta-analysis [47]

## 3. Results

### 3.1. Identification and Selection of Studies

Figure 1 shows the flowchart of the literature search according to PRISMA guidelines. Initially, we identified a total of 10,354 records. After removal of duplicates, 9906 records were left. Then, we reviewed 21 records with relevant titles and abstracts. Finally, we included eight original research studies in our review and meta-analysis [36,41,42,48,49,50,51,52].

### 3.2. Characteristics of the Studies

Table 1 shows the main characteristics of the eight studies included in our review. A total of 5754 nurses were included in our review and meta-analysis. The sample size in the included studies ranged from 242 nurses to 1761. Two studies had been conducted in Europe [48,51], two studies in China [41,42], two studies in Australia [49,52], one study in the USA [50], and one study in Egypt [36]. All studies were cross-sectional and had been conducted after 2010. The percentage of female nurses ranged from 79.0% to 96.6%. Seven studies used convenience samples while one study used a purposive sample [42]. All studies included nurses working in hospitals. Seven studies used the Survey of Perceived Organizational Support to measure organizational support while one study used the Perceived Organizational Support—Simplified Version Scale [42]. Three studies used the Turnover Intention Scale to measure turnover intention [36,41,42] while five studies used other self-developed scales that have been validated [48,49,50,51,52]. The response rate among studies ranged from 21.4% to 96.3%.

### 3.3. Quality Assessment

Appendix A shows the quality of the studies included in our review. Quality was moderate in five studies [41,48,49,51,52] and good in three studies [36,42,50]. Failure to identify and eliminate confounding factors was the main threat to study quality.

### 3.4. Meta-Analysis

All studies reported a correlation coefficient between organizational support and turnover intention among nurses. The correlation coefficients and *p*-values for all studies are shown in Table 1. All studies found negative correlations between organizational support and turnover intention ranging from −0.10 [36] to −0.51 [50]. We found a statistically significant negative correlation since the pooled correlation coefficient was −0.32 (95% CI: −0.42 to −0.21, *p* < 0.001) (Figure 2). The overall correlation coefficient suggested a moderate negative correlation between organizational support and turnover intention. Heterogeneity between results was high (I^2^ = 93%, *p*-value for the Hedges Q statistic < 0.001).

A leave-one-out sensitivity analysis showed that our results were stable when each study was excluded. In particular, the pooled correlation coefficient varied between −0.29 (95% CI: −0.41 to −0.16, *p* < 0.001), with Liu et al. [42] excluded, and −0.35 (95% CI: −0.46 to −0.23, *p* < 0.001), with Galletta et al. [51] excluded.

Egger’s test (Egger bias = −1.43, 95% CI: −6.56 to 3.70, *p* = 0.63) and funnel plot (Figure 3) suggested the absence of publication bias in the eight studies.

Subgroup analysis showed that the negative correlation between organizational support and turnover intention was stronger in studies in China (pooled r = −0.33, 95% CI: −0.81 to 0.41, I^2^ = 82%) and Australia (pooled r = −0.33, 95% CI: −0.87 to 0.57, I^2^ = 75%) than in studies in Europe (pooled r = −0.25, 95% CI: −0.76 to 0.45, I^2^ = 75%). Moreover, the negative correlation was stronger for studies with a low risk of bias (pooled r = −0.34, 95% CI: −0.73 to 0.22, I^2^ = 97%) than studies with a moderate risk of bias (pooled r = −0.28, 95% CI: −0.37 to −0.19, I^2^ = 76%).

Meta-regression showed that the pooled correlation coefficient was independent of the percentage of females (beta = −0.01, *p* = 0.11), data collection time (beta = 0.01, *p* = 0.65), sample size (beta = −0.00004, *p* = 0.73), and response rate (beta = 0.002, *p* = 0.19).

Only two studies had conducted multivariable analysis to estimate the independent effect of organizational support on nurses’ turnover intention. Both of these studies found a negative association between organizational support and turnover intention. However, Liu et al. [42] found a statistically significant association (unstandardized coefficient beta = −0.012, *p* < 0.01) while Filipova [50] did not find a statistically significant association (unstandardized coefficient beta = −0.010, *p* > 0.05).

Moreover, only two studies investigated potential mediators of the relationship between organizational support and turnover intention. In particular, Filipova [50] found that organizational commitment completely mediated the negative relationship between organizational support and turnover intention while job satisfaction partially mediated this relationship. In a similar way, Shacklock et al. [52] found that job satisfaction partially mediated the negative relationship between organizational support and turnover intention. No studies investigated potential moderators of the relationship between organizational support and turnover intention. 

## 4. Discussion

Our meta-analysis highlighted the moderate negative correlation between organizational support and turnover intention. Moreover, two studies found a negative association between organizational support and turnover intention after eliminating confounding [42,50]. Nurses’ turnover intention has a significant impact on the functioning of healthcare organizations. This impact includes the understaffing of nursing departments from which nurses leave, the negative impact on nurses’ mental health, the deterioration of patient safety (falls and medical errors), and patients’ dissatisfaction with the healthcare services provided [53]. The impact also includes the waste of financial and other resources in recruiting new staff and training them to fully assume their duties [53,54]. As there are already serious safety problems in the provision of healthcare [55] and issues with the mental health of nurses [56], nurses’ turnover intention seems to exacerbate the existing situation.

The decision of nurses to leave the profession is not a sudden decision but a process that goes through three stages [57]. In the first stage, the psychological, the employee through turnover-intention psychological responses to negative aspects of organization or job. He/she begins to feel dissatisfaction with his/her job, showing reduced commitment and attachment to his/her organization. In the second stage, the cognitive, turnover intention is defined as the final cognitive step leading to actual turnover. In the third stage, the behavioral, the employee now changes his/her behavior as, in addition to expressing his/her desire to leave, he/she loses enthusiasm and is late to work or even absent. A recent study in Greece, involving 629 nurses, showed that 60.9% of nurses choose quiet quitting [26]. Employees who choose this behavior reduce their effort, perform only highly necessary tasks, do not propose new ideas and practices, do not stay overtime, and do not come to work early. Their goal is to work only as much as necessary to avoid being fired. The study showed that nurses who opt for quiet quitting, in which they reduce their performance at work, are more likely to have high levels of turnover intention. Therefore, the factor that triggers turnover intention is the working environment of nurses, and the management of healthcare organizations should focus on improving it.

Even if nurses are dissatisfied with their work or experience burnout and report their turnover intention, organizational support can mitigate the effect of the two factors mentioned above on their turnover intention. Four studies showed the indirect, mediating role of organizational support on turnover intention through job satisfaction [42,49,51,52] and two studies showed the similar role through burnout [42,48]. Nurses report moderate levels of job satisfaction in primary healthcare settings and high levels of dissatisfaction in secondary ones [58,59]. Even now, in the post-COVID-19 era, as the workload has been reduced and the functioning of healthcare organizations has been normalized, nurses continue to show dissatisfaction at a higher rate than other healthcare professionals [23]. When nurses report increased satisfaction with their work, the likelihood of turnover intention is reduced [60]. The main organizational factor associated with increased job satisfaction is a good working environment, characterized by well-staffed nurses, adequate resources, reduced workload, satisfactory salaries and rewards, opportunities for development and promotion, recognition of the role of nurses, and effective supervision [61,62]. The aforementioned factors constitute the conceptual framework of perceived organizational support [30,63]. Therefore, ensuring and improving these factors constitutes a strong organizational support for nurses, which is directly linked to increasing their job satisfaction, and will indirectly reduce their turnover intention. In addition to job dissatisfaction, nurses also experience high rates of burnout. Before the COVID-19 pandemic, it is estimated that one out of three nurses reported being exhausted [64], and this rate increased to very high levels after the pandemic and its impact [23]. Burnout appears to be a strong predictor of nurses’ turnover intention [65,66]. The effect of the way that nurses’ work environments are organized and operated is also related to their burnout, in addition to their dissatisfaction. Factors such as low/inadequate nurse staffing levels, ≥ 12 h shifts, low autonomy, poor nurse–physician relationships, poor supervisor/leader support, job insecurity, and reduced opportunities for nurses to participate in hospital affairs make up the organizational factors that lead nurses to burnout [67,68]. During the COVID-19 pandemic, the extremely difficult and demanding working conditions combined with the organizational inefficiencies of the past resulted in a large proportion of nurses becoming burnt out [9]. Recognition of nurses’ work, opportunities for development, and ensuring good working conditions through ongoing organizational support reduce nurses’ burnout [69], increase trust in the organization [48,69], and ultimately reduce the chances of nurses’ turnover intention being actualized [48].

Among studies in our review, Sheng et al. [41] found that high organizational support plays a mediating role in the relationship between nurses’ practice environment and their well-being, which in turn is correlated to turnover intention [41]. The working environment and the demanding nature of nurses’ work negatively affect their well-being [70], resulting in high rates of anxiety, depression, psychological stress, and post-traumatic stress disorder [71,72,73]. The more the well-being of nurses deteriorates, the higher the likelihood of turnover intention becomes [20,74]. Nurses often feel both weak and defenseless in the face of difficult situations, as in the case of the COVID-19 pandemic [75]. In these challenging and difficult times, the support they receive, either at the departmental level from their supervisors or at the organizational level, helps them cope with these difficulties and mitigates their impact on their well-being [75,76].

Additionally, Liu et al. found that violence in the nurses’ workplace is a factor associated with an increase in turnover intention, while organizational support mediates the effect of violence on turnover intention [42]. Incidents of violence, both physical and verbal, have a high impact on nursing staff, with nurses in emergency departments almost all reporting being victims of violence [77,78]. Nurses are dissatisfied with their organization in terms of the prevention and management of violent incidents as well as their lack of training in dealing with such incidents [79]. The consequences of violent incidents affect the quality of care, employee performance, and nurses’ mental health and the willingness to leave their jobs [78,80]. The consequences of violence, which even can lead to serious physical injury and death, make it imperative to protect nurses, who feel defenseless and vulnerable. When nurses receive organizational support and feel less vulnerability, their desire to leave their jobs is mitigated [81].

Moreover, Filipova found that increased nurses’ commitment mediates the relationship between perceived organizational support and their intent to leave their jobs [50]. High commitment is an important factor influencing the quality of care and hospital performance [82]. A significant number of factors have been found to affect nurses’ commitment, e.g., well-being, satisfaction, leadership and management style, and behavior and working environment [83]. When organizational support is low and nurses wish to leave their jobs, then, through organizational commitment, the negative effect of support on turnover intention is mitigated [84].

In summary, organizational support both directly and indirectly influences nurses’ turnover intention. Through recognizing nurses’ work, ensuring that nurses have the resources needed to provide care, and providing rewards and opportunities for improvement, the likelihood of nurses declaring their turnover intention is reduced. In particular, the management of healthcare organizations should aim to ensure adequate nursing staffing, monetary rewards, opportunities for promotion, and support from the supervisor. Also, organizational support reduces burnout and increases nurses’ satisfaction. As nurses’ satisfaction increases and burnout decreases, the percentage of nurses who report turnover intention decreases.

As the issue of turnover intention is complex, the management of healthcare organizations should also take into account and manage the other factors that can lead to turnover intention. These factors include job stress and fatigue, burnout, depression, organizational justice and culture, job prospect and stability, relationships with managers and colleagues, and the work environment [17,18,20,21,22,85].

## 5. Limitations

Our study has several limitations. First, the number of studies included in our review and meta-analysis is small. Moreover, the number of studies for subgroup analyses is even smaller. For example, there is only one study set in the USA and one study in Africa. Thus, the representativeness of our results is limited. Further studies in different countries, cultures, and settings should be conducted to obtain more valid results. Second, only two studies assessed the independent effect of organizational support on turnover intention in nurses by applying multivariable models. All studies estimated the correlation between organizational support and turnover intention. Thus, future studies should employ multivariable models to eliminate confounding in the relationship between organizational support and turnover intention. Moreover, we suggest that scholars explore the role of mediating or/and moderating variables since structural equation modeling enables us to perform mediation/moderation analysis in a valid way. Third, all studies included in our review were cross-sectional, and a causal relationship between organizational support and turnover intention cannot be established. Measuring organizational support and turnover intention at the same time may produce a spurious correlation. Thus, there is a need for longitudinal studies, which can further explain the relationship between organizational support and turnover intention. Fourth, seven studies used convenience samples, and only one study used a purposive sample. For example, nurses in all studies were mainly females. Therefore, selection bias is potential in our review. Further studies with more representative and stratified samples can add valuable evidence. Finally, we searched six major databases, applying the guidelines for systematic reviews, but it is still possible for us to have missed studies in our evaluation. For example, we did not include studies in non-English languages and grey literature.

## 6. Conclusions

Our meta-analysis suggests a moderate negative correlation between organizational support and turnover intention in nurses. In other words, nurses who have experienced more organizational support tend to be less likely to leave their jobs than those who have experienced less organizational support. As nursing understaffing characterizes a significant number of healthcare organizations and the shortage of nurses is a constant threat to health systems, the turnover intention of nurses may exacerbate the existing situation. This study highlighted the direct and indirect association between organizational support and turnover intention and, also, the specific characteristics of organizational support that should be strengthened by the administrations of healthcare organizations to reduce turnover intention. Our findings constitute an alarm for organizations, policy makers, and nursing managers to pay more attention to organizational support.

## Figures and Tables

**Figure 1 healthcare-12-00291-f001:**
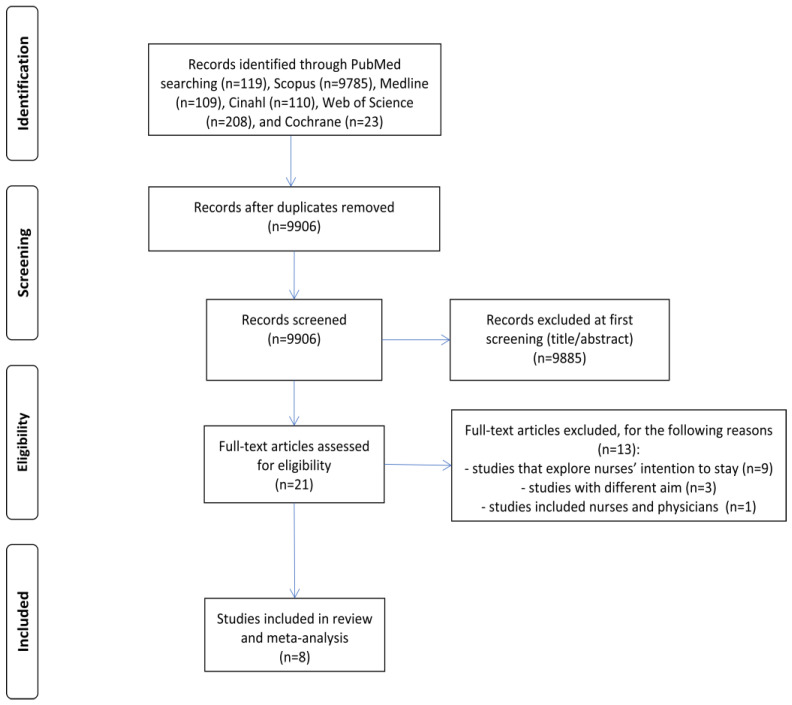
Flowchart of the systematic review.

**Figure 2 healthcare-12-00291-f002:**
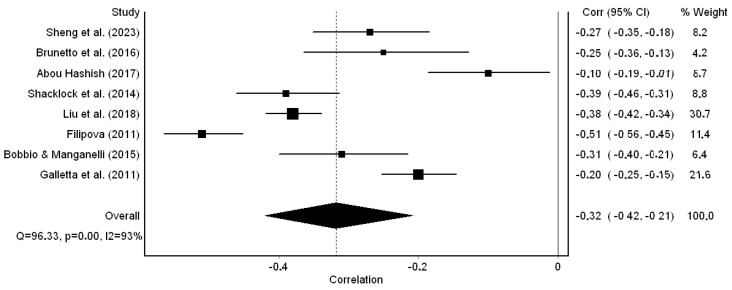
Forest plot of the eight studies included in this meta-analysis [36,41,42,48,49,50,51,52].

**Figure 3 healthcare-12-00291-f003:**
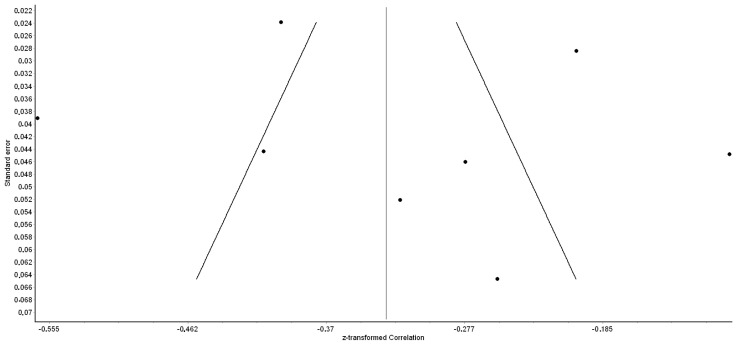
Funnel plot of the correlation coefficient between organizational support and turnover intention among nurses.

**Table 1 healthcare-12-00291-t001:** Main characteristics of studies included in this systematic review.

Reference	Country	Data Collection Time	Females (%)	Age, Mean (SD)	Sample Size (n)	Study Design	Sampling Method	Clinical Settings	Assessment Tool for Organizational Support	Assessment Tool for Turnover Intention	Response Rate (%)	Correlation Coefficient (*p*-Value)
(Sheng et al., 2023) [41]	China	2020–2021	96.2	27.0 (3.9)	474	Cross-sectional	Convenience	Hospitals	SPOS	TIS	96.3	−0.27 (<0.01)
(Brunetto et al., 2016) [49]	Australia	2013	83.2	41–60 years: 64.8%	242	Cross-sectional	Convenience	Hospitals	SPOS	Eight-item scale	33.0	−0.25 (<0.01)
(Abou Hashish, 2017) [36]	Egypt	NR	NR	≤29 years: 47.2%; 30–40: 33.0%; ≥41: 19.8%	500	Cross-sectional	Convenience	Hospitals	SPOS	TIS	78.5	−0.10 (0.16)
(Shacklock et al., 2014) [52]	Australia	2010–2011	93.7	46.5 (10.4)	510	Cross-sectional	Convenience	Hospitals	SPOS	Three-item scale	31.5	−0.39 (<0.001)
(Liu et al., 2018) [42]	China	2016–2017	96.6	≤30 years: 51.1%; 31–50: 45.0%; ≥51: 3.9%	1761	Cross-sectional	Purposive	Hospitals	POS-SVS	TIS	85.2	−0.38 (<0.001)
(Filipova, 2011) [50]	USA	2010	94.0	44–53 years: 37.0%	656	Cross-sectional	Convenience	Hospitals	SPOS	Three-item scale	21.4	−0.51 (<0.001)
(Bobbio & Manganelli, 2015) [48]	Italy	2012	79.0	42.3 (8.1)	371	Cross-sectional	Convenience	Hospitals	SPOS	Three-item scale	41.0	−0.31 (<0.01)
(Galletta et al., 2011) [51]	France	2010	81.5	37.0 (7.9)	1240	Cross-sectional	Convenience	Hospitals	SPOS	Two-item scale	64.0	−0.20 (<0.001)

NR: not reported; POS-SVS: Perceived Organizational Support—Simplified Version Scale; SD: standard deviation; SPOS: Survey of Perceived Organizational Support; TIS: Turnover Intention Scale.

## Data Availability

Our data are available from the corresponding author on reasonable request.

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
