# Peer review of "Association between Organizational Support and Turnover Intention in Nurses: A Systematic Review and Meta-Analysis"

_healthcare, 2024, doi:10.3390/healthcare12030291_

Round 1

Reviewer 1 Report

Comments and Suggestions for Authors

Ioanna V. Papathanasiou submitted to Healthcare an article focusing to a systematic review and meta-analysis on the association between the organizational support and turnover intention in nurses. 

The manuscript is well structured and is useful for experts in the field. The methodology used is clear and compliant with the common standard for reviews.

I suggest that the authors improve the discussions section, providing as take-home messages a synoptic overview of the best organizational evidence found, to be shared with colleagues involved in decision-making processes and management, to improve the working conditions of nurses in the numerous occupational contexts in which they find employment.

Comments on the Quality of English Language

Minor editing of English language required

Author Response

Dear Reviewer 1

Thank you very much for the peer review of the paper “Association between organizational support and turnover intention in nurses: a systematic review and meta-analysis” and your comments, which have improved the quality of the manuscript.

We have addressed all the comments (highlighted in yellow) in the revised text. Please, find below an item-by-item answer to your comments. Also, we made changes in the manuscript according to the other Reviewers’ instructions.

Hoping the revised manuscript fulfils the journal’s standards, we thank you for your courtesy.

We are looking forward to your response.

Yours sincerely,

The authors

Comment

I suggest that the authors improve the discussions section, providing as take-home messages a synoptic overview of the best organizational evidence found, to be shared with colleagues involved in decision-making processes and management, to improve the working conditions of nurses in the numerous occupational contexts in which they find employment.

Answer: Done

We added the following text at the end of the discussion section:

In summary, organizational support both directly and indirectly influences nurses' turnover intention. Through recognizing nurses' work, ensuring that nurses have the re-sources needed to provide care, and providing rewards and opportunities for improvement, the likelihood of nurses declaring turnover intention is reduced. In particular, the management of health care organizations should aim to ensure adequate nursing staffing, monetary rewards, opportunities for promotion and support from the supervisor. Also, organizational support reduces burnout and increases nurses' satisfaction. As nurses' satisfaction increases and burnout decreases, the percentage of nurses who report turnover intention decreases.

Reviewer 2 Report

Comments and Suggestions for Authors

The abstract of this manuscript would benefit from greater conciseness and focus. It currently includes details that are more suited to the general methodology section, such as procedural elements like “We searched…” Such information could be more appropriately detailed in the main body of the text.

Regarding the concept of organizational support, the manuscript currently presents it as a broad term. However, the implications and manifestations of organizational support can vary significantly across different organizations and countries. This diversity needs to be acknowledged and addressed. I recommend that the author provides a more nuanced and specific definition of organizational support, especially as it functions as one of the predictor variables in the Meta-Analysis. This will enhance the manuscript's relevance and applicability to a global audience.

In the context of nurse turnover, the manuscript seems to overlook the complexity of this issue. Nurse turnover is not only a result of job dissatisfaction but can also stem from burnout, depression, and a variety of demographic and cultural factors. These elements should be considered and integrated into the analysis to present a more comprehensive understanding of the issue.

Finally, the manuscript's reliance on simple correlational analysis between two variables to predict nurse turnover is a significant limitation. This approach fails to consider potential mediators or moderators that could influence the relationship between these variables. For a more robust and informative analysis, I suggest incorporating additional statistical methods that can account for these mediating and moderating factors. This will provide a more comprehensive and realistic framework for understanding the dynamics at play in nurse turnover.

In summary, while the manuscript provides valuable insights, it could greatly benefit from a more detailed, nuanced, and multifaceted approach to its subject matter.

Author Response

Dear Reviewer 2

Thank you very much for the peer review of the paper “Association between organizational support and turnover intention in nurses: a systematic review and meta-analysis” and your comments, which have improved the quality of the manuscript.

We have addressed all the comments (highlighted in yellow) in the revised text. Please, find below an item-by-item answer to your comments. Also, we made changes in the manuscript according to the other Reviewers’ instructions.

Hoping the revised manuscript fulfils the journal’s standards, we thank you for your courtesy.

We are looking forward to your response.

Yours sincerely,

The authors

Comment

The abstract of this manuscript would benefit from greater conciseness and focus. It currently includes details that are more suited to the general methodology section, such as procedural elements like “We searched…” Such information could be more appropriately detailed in the main body of the text.

Answer: Done

We remove details regarding our methods. In particular, we remove the following text from the Abstract:

….A systematic review and meta-analysis were performed. We searched PubMed, Medline, Scopus, Cinahl, Web of Science, and Cochrane from inception to July 21, 2023. Heterogeneity between results was high. Thus, we applied a random effect model to estimate pooled correlation coefficient between organizational support and turnover intention…

….Our inclusion criteria were the following: (a) studies that included nurses working in clinical settings, (b) articles published in English, (c) studies that investigated the relationship between organizational support and turnover intention in nurses, and (d) studies that used valid instruments to measure organizational support and turnover intention…

Comment

Regarding the concept of organizational support, the manuscript currently presents it as a broad term. However, the implications and manifestations of organizational support can vary significantly across different organizations and countries. This diversity needs to be acknowledged and addressed. I recommend that the author provides a more nuanced and specific definition of organizational support, especially as it functions as one of the predictor variables in the Meta-Analysis. This will enhance the manuscript's relevance and applicability to a global audience.

Answer: Done

Dear Reviewer, thank you especially for this comment. We add the following text in the section 2.2. Selection process:

Organizational support is a broad term that can vary across different organizations and countries. In our review we included studies that measured the perceived organizational support among nurses. In particular, perceived organizational support was defined as nurses’ overall perceptions and beliefs about how much the organizations value and respect nurses’ well-being and job satisfaction.

Comment

In the context of nurse turnover, the manuscript seems to overlook the complexity of this issue. Nurse turnover is not only a result of job dissatisfaction but can also stem from burnout, depression, and a variety of demographic and cultural factors. These elements should be considered and integrated into the analysis to present a more comprehensive understanding of the issue.

Answer: Done

In the introduction section, lines 56-61, we present the most important factors leading to turnover intention. These factors include burnout, depression, job stress and some work-related factors.

Also, at the end of the discussion section, we added the following text:

As the issue of turnover intention is complex, the management of healthcare organizations should also take into account and manage the other factors that can lead to turnover intention. These factors include job stress and fatigue, burnout, depression, organizational justice and culture, job prospect and stability, relationships with managers and colleagues, and work environment [17,18, 20-22, 85].

Comment

Finally, the manuscript's reliance on simple correlational analysis between two variables to predict nurse turnover is a significant limitation. This approach fails to consider potential mediators or moderators that could influence the relationship between these variables. For a more robust and informative analysis, I suggest incorporating additional statistical methods that can account for these mediating and moderating factors. This will provide a more comprehensive and realistic framework for understanding the dynamics at play in nurse turnover.

Answer: Done

Dear Reviewer, thank you especially for this comment. Two studies investigated potentials mediators into the relationship between organizational support and turnover intention, while no studies investigated potentials moderators. Since data are limited, we cannot perform any statistical methods. However, we add this valuable information into the Results section as follows:

….Moreover, only two studies investigated potentials mediators into the relationship between organizational support and turnover intention. In particular, Filipova [50] found that organizational commitment completely mediated the negative relationship between organizational support and turnover intention, while job satisfaction partially mediated this relationship. In a similar way, Shacklock et al. [52] found that job satisfaction was a partially mediation in the negative relationship between organizational support and turnover intention. No studies investigated potentials moderators into the relationship between organizational support and turnover intention

Moreover, we add the following comment on the Limitations section:

Moreover, we suggest scholars to explore the role of mediating or/and moderating variables since the structural equation modeling enable us to perform mediation/moderation analysis in a valid way

Reviewer 3 Report

Comments and Suggestions for Authors

Improvement proposals

In the introduction - the research objective must be stated, as well as the questions that this study seeks to answer.

After the introduction, the literature review must be placed, where all relevant literature for understanding this study must be placed.

Line 107 – “We searched PubMed, Medline, Scopus, Cinahl, Web of Science, and Cochrane from inception to August 21, 2023. Authors must mention the beginning and end of the search.

Line 300 – „Moreover, two studies found a negative association between organizational support and turnover intention after eliminating confounding.“ Authors must mention, specifically, what these studies are.

Line 318 – “A recent study showed that nurses who opt for quiet quitting, in which they reduce their performance at work, are more likely to have high levels of turnover intention [26]. Authors must specifically mention this study.

Line 415 – „Our findings are an alarm for organizations, policy makers and nursing managers to pay more attention to organizational support. Authors must specifically mention how the organization can contribute, support and improve its support for nurses.

The authors must describe the main contribution of this study to the understanding of this topic, as well as include future studies that can contribute to a more complete understanding of the topic.

Author Response

Dear Reviewer 3

Thank you very much for the peer review of the paper “Association between organizational support and turnover intention in nurses: a systematic review and meta-analysis” and your comments, which have improved the quality of the manuscript.

We have addressed all the comments (highlighted in yellow) in the revised text. Please, find below an item-by-item answer to your comments. Also, we made changes in the manuscript according to the other Reviewers’ instructions.

Hoping the revised manuscript fulfils the journal’s standards, we thank you for your courtesy.

We are looking forward to your response.

Yours sincerely,

The authors

Comment

In the introduction - the research objective must be stated, as well as the questions that this study seeks to answer.

Answer: Done

In the introduction section, lines 93-95, we stated the aim of the study.

Comment

After the introduction, the literature review must be placed, where all relevant literature for understanding this study must be placed.

Answer: According to the journal's guidelines and the specific template on which the article must be written, no other section, such as the literature review, is provided.

After the first paragraph of the introduction, lines 38-49, the literature review actually begins.

Comment

Line 107 – “We searched PubMed, Medline, Scopus, Cinahl, Web of Science, and Cochrane from inception to August 21, 2023. Authors must mention the beginning and end of the search.

Answer: Done

We added the following text in section "2.1. Search methods", lines 103-104:

The duration of the literature search of the studies by the authors lasted from 14 to 21 August 2023.

Comment

Line 300 – „Moreover, two studies found a negative association between organizational support and turnover intention after eliminating confounding.“ Authors must mention, specifically, what these studies are.

Answer: Done

We added the citations of the two studies, line 307.

Comment

Line 318 – “A recent study showed that nurses who opt for quiet quitting, in which they reduce their performance at work, are more likely to have high levels of turnover intention [26]. Authors must specifically mention this study.

Answer: Done

We added the following text in discussion section, lines 323-327:

A recent study in Greece, involving 629 nurses, showed that 60.9% of nurses choose quiet quitting [26]. Employees who choose this behavior reduce their effort, perform only highly necessary tasks, do not propose new ideas and practices, do not stay overtime, and do not come to work early. Their goal is to work only as much as necessary to avoid being fired.

Comment

Line 415 – „Our findings are an alarm for organizations, policy makers and nursing managers to pay more attention to organizational support. Authors must specifically mention how the organization can contribute, support and improve its support for nurses.

Answer: Done

We added the following text in discussion section, lines 396-404:

In summary, organizational support both directly and indirectly influences nurses' turnover intention. Through recognizing nurses' work, ensuring that nurses have the re-sources needed to provide care, and providing rewards and opportunities for improvement, the likelihood of nurses declaring turnover intention is reduced. In particular, the management of health care organizations should aim to ensure adequate nursing staffing, monetary rewards, opportunities for promotion and support from the supervisor. Also, organizational support reduces burnout and increases nurses' satisfaction. As nurses' satisfaction increases and burnout decreases, the percentage of nurses who report turnover intention decreases.

Comment

The authors must describe the main contribution of this study to the understanding of this topic, as well as include future studies that can contribute to a more complete understanding of the topic.

Answer: Done

We added the following text in conclusion section, lines 438-444:

As nursing understaffing characterizes a significant number of health care organizations and the shortage of nurses is a constant threat to health systems, the turnover intention of nurses may exacerbate the existing situation. This study highlighted the direct and indirect association between organizational support and turnover intention and also the specific characteristics of organizational support that should be strengthened by the administrations of health care organizations to reduce turnover intention.

Regarding the need for future studies, in the limitation section, lines 424-427 and 429-430, we stated:

….. Measuring organizational support and turnover intention at the same time may produce a spurious correlation. Thus, there is a need for longitudinal studies, which can further ex-plain the relationship between organizational support and turnover intention.

……… Further studies with more representative and stratified samples can add valuable evidence.

Round 2

Reviewer 2 Report

Comments and Suggestions for Authors

I accept revision by the authors

Reviewer 3 Report

Comments and Suggestions for Authors The article was improved with the requested suggestions